# Codenames as a Game of Co-occurrence Counting

**Réka Cserháti** and **István Kolláth** and **András Kicsi** and **Gábor Berend**
Institute of Informatics
University of Szeged
{cserhatir,kollathistvan}@gmail.com
{akicsi,berendg}@inf.u-szeged.hu

## Abstract

Codenames is a popular board game, in which knowledge and cooperation between players play an important role. The task of a player playing as a spymaster is to find words (clues) that a teammate finds related to as many of some given words as possible, but not to other specified words. This is a hard challenge even with today's advanced language technology methods.

In our study, we create spymaster agents using four types of relatedness measures that require only a raw text corpus to produce. These include newly introduced ones based on co-occurrences, which outperform FastText cosine similarity on gold standard relatedness data. To generate clues in Codenames, we combine relatedness measures with four different scoring functions, for two languages, English and Hungarian. For testing, we collect decisions of human guesser players in an online game, and our configurations outperform previous agents among methods using raw corpora only.

## 1 Introduction

One of the central subjects of artificial intelligence research has long been the development of agents that play various games at the human level or better. Most studies in the field focus on combinatorial games, that can be easily formalized mathematically, such as chess and go (see, for example, Allis et al., 1994). The popular board game Codenames is different from these in many aspects and may provide an excellent experimental ground in areas such as predicting human behavior or implementing human-machine cooperation.

In the original game, two teams compete against each other. A board of 25 word cards contains cards belonging to the blue or red team, neutral cards, and an instant defeat card (black). A team wins if all cards of their team are revealed earlier than the cards of the other team, or if the opponent reveals the black card. However, only one person (the spymaster) from both teams knows which card is of what color. Therefore, the spymasters give the team a clue each turn, which consists of a clue word and a number. The other members of the team (guessers), in consultation with each other, reveal cards on the board they think are related to the clue word, until they bet on a wrong card, or reach the limit given by the spymaster as a number.

This means it is possible to create two types of agents for the game, spymasters and guessers. The main task of both agents is to be able to cooperate with human players. To create agents capable of such high-level cooperation, we need to be able to predict human behavior in the game. This task includes modeling the relatedness of words, with the aim of obtaining relatedness measures that represent human perception well.

This task is highly related to word association modeling, which has been studied extensively in psycholinguistics for a long time (Palermo and Jenkins, 1964; McNeill, 1966), but is by no means equivalent to it. In word association experiments, subjects should name any word associated with a given word as quickly as possible, but in this case, the spymaster's task is to find a word that is related to as many words from a given set as possible, but not or significantly less closely to a set of other words. The time allotted for the task is also limited at most very loosely (by the patience of the other players), and based on personal experiences, spymasters often use several minutes of thinking time to come up with the right clue. For this reason, connected words are often related in a complex way, even indirectly. The task of agents – to find words in the table related to the clue word – is more like simple associations, but time is not dominant here either, and more complex, indirect relations also matter. In a game between people, the relationship and common knowledge between the players can also count, but this is not an influencing factor in a game with an agent.

## 2  A Mathematical Model of the Game

Suppose that for a dictionary $V$, a similarity matrix $S \in \mathbb{R}^{|V| \times |V|}$ exists in which $S_{ij} = s(w_i, w_j)$ is the exact measure of the relationship between any two words $w_i, w_j$, that is, the relationships are just as strong according to every person. Then the implementation of the guesser agent is simple: from the words on the board, always choose the one that is most closely related to the clue word. This way, a greedy spy-agent is also simple: let $v_i$ be the $i$-th word of the dictionary, and for every $i$, let $[w_{i1}, w_{i2}, ..., w_{in}]$ be the unrevealed words on the board, ordered by the relatedness to $v_i$, from the most closely related to the least related one. Then we look for $i$ for which the largest number $k$ exists, such that $w_{i1}, w_{i2}, ..., w_{ik}$ are all words belonging to the agent's team. Then $v_i$ will be the clue word, and $k$ the number of targeted words.

However, under such conditions, the behavior of the guessers is deterministic, which means the two spymasters are playing against each other. The dictionary, that is, the number of their possible decisions is finite, and spymasters know the outcome of each decision, which means they know each other's possible strategies. Thus, the game becomes a sequential game with perfect information, like e.g. chess, go, or tic-tac-toe. A greedy decision is not necessarily optimal, since a spymaster needs to consider what options they will have later, depending on their own and the other spymasters' decisions, and should optimize their move based on that. Within such a framework, the development of an optimal strategy may be the subject of further research, but is no more connected to computational and cognitive linguistics, so we will not discuss this further in this article.

The above conditions are, of course, far from reality, since such a distance function, which perfectly corresponds to the mental representations of all people, certainly does not exist. This is clear from the fact that in classical association tests, where the actual task is to find nearest neighbors, the subjects never give the same answer (Palermo and Jenkins, 1964; Postman and Keppel, 2014). However, it is a meaningful task to create a similarity function and construct a similarity matrix $S \in \mathbb{R}^{V \times V}$, in which $S_{ij} = s(w_i, w_j)$ approximates the average similarity perceived by people.

Furthermore, based on the similarity approximations, we can define a scoring function for possible clues, which realistically ranks them according to how many correct guesses a human guesser player is expected to give. Our distance matrix and our scoring function together determine a greedy spymaster agent. Since this task is challenging in itself, we disregard the possible non-greedy strategies and focus on optimizing similarity approximations and clue scoring functions for one round only.

## 3  Related Work

### 3.1  Associations

Word associations have been a subject of active research for a long time in cognitive science and psycholinguistics for various reasons. They were used to study mental functioning, memory, and certain diseases. Word associations were also applied for modeling the cognitive lexicon and some linguistic processes (summarized by Bel-Enguix et al., 2019).

One can create a graph (Bel-Enguix, 2014), and its transformation to a word embedding model (Bel-Enguix et al., 2019), specifically for modeling associations, but these require difficult-to-obtain association data. This would be a high resource requirement and would make it difficult to apply such methods in various languages.

Instead, we can use methods that require only raw corpora. For this, the results of Spence and Owens (1990) are the most important studies of associations. They have shown that the amount of co-occurrences of words in a corpus is a good indicator of the semantic relationship between them and is also suitable for measuring the strength of associations. Bel Enguix et al. (2014) also predict associations from co-occurrences, using a network of bigram counts. Similar to their methods, we use weighted co-occurrences explicitly to model the connection of words (for details, see 4.1.).

### 3.2  Language graphs

Although the canonical way to represent words is to assign them to vectors, if the goal is to model connections between words, a graph structure is at least as suitable. When each word is represented by a vector, the similarity between them is most often calculated as the cosine of the angle of the two vectors. In the case of graph representations, all words in the dictionary correspond to the vertices of a large graph, and the distance between them can be defined in many ways depending on the graph. One option is the length or weight of the shortest path between the two vertices. Knowledge graphs

(Miller, 1992; Speer and Havasi, 2012; Navigli and Ponzetto, 2010a) were already used to model word connections in previous Codenames agents, but other types of language graphs also exist, which could be utilized for this task as well.

Hope and Keller (2013), for example, use a graph of co-occurrences for word sense induction. Later Pelevina et al. (2016) use a similar method to disambiguate word embedding models.

Another graph, created as an alternative for word embeddings, is GraphGlove (Ryabinin et al., 2020), where the edges of the graph are optimized by the cost function of GloVe (Pennington et al., 2014b), so that the shortest path between two vertices gives the distance of the corresponding words.

### 3.3 Codenames agents

To the best of our knowledge, the first algorithms similar to Codenames agents have been created by Shen et al. (2018) specifically to model human associations. In their simplified game, the board always consists of three nouns, and the agent gives a clue that must be one of three adjectives, and refers to exactly two of the board words. Their clues were generated based on the following five similarity functions:

- probability of bigrams relative to word frequency,
- cosine similarity in Skip-gram (Mikolov et al., 2013),
- cosine similarity in GloVe (Pennington et al., 2014a),
- connection according to the knowledge graph ConceptNet5 (Speer and Havasi, 2012),
- similarity in topic modeling.

They found that the behavior of human players is best modeled on the probabilities of bigrams, which is in line with the results of (Spence and Owens, 1990) (although the latter calculated co-occurrences with much larger window size).

Kim et al. (2019) were the first to build agents designed explicitly to play the game. As a background to their relatedness measure, they used

- CBOW, Skip-gram and GloVe word embeddings (in multiple configurations),
- and the WordNet database (Miller, 1992) with a number of different distance functions.

However, in their study, they do not evaluate the performance of agents with human data, but by pair-ing spymaster and guesser agents, which reveals only the similarity of the two agents, regardless of their ability to interact with humans.

Jaramillo et al. (2020) calculated similarity functions from the following representations:

- TF-IDF similarity calculated from Wikipedia articles and dictionary definitions,
- a naive-Bayesian classification of words, and
- word embeddings extracted from the first layer of the GPT2 language model (Radford et al., 2019).

Of these methods, they find GPT2 vectors best suited to model word relatedness.

The latest article on the topic is (Koyyalagunta et al., 2021), in which, in addition to the previously used Skip-gram and GloVe word embeddings, to produce their similarity matrices they use

- FastText (Bojanowski et al., 2017),
- the BERT model (Devlin et al., 2018),
- and the BabelNet knowledge graph (Navigli and Ponzetto, 2010b), with a framework that associates words according to special rules, developed specifically for this purpose.

In addition to calculating the relatedness between words, the above works also differ in the scoring functions of the possible clues. Without limiting the generality, we assume that our agent plays in the blue team, that is, our clues refer to the blue words. Using the notations of Koyyalagunta et al. (2021), let $\tilde{c}$ be a possible clue word, $I_n$ a set of targeted (intended) words, that is, the $n$ closest blue words to $\tilde{c}$, $R$ the set of all bad words that do not belong to the team (red words), and $s(\cdot, \cdot)$ a function that calculates the similarity or relatedness of two words. The scoring function of Kim et al. (2019) is then

$$g_{Kim}(\tilde{c}, n) = \begin{cases} \min_{b \in I_n} s(\tilde{c}, b), \\ \quad \text{if } \min_{b \in I_n} s(\tilde{c}, b) > \max_{r \in R} s(\tilde{c}, r) \\ 0, \text{ otherwise.} \end{cases}$$

(1)

Jaramillo et al. (2020) takes the same function, but adds penalties based on the color of the cards. Koyyalagunta et al. (2021), on the other hand, define another scoring function:

$$g_{Koyy}(\tilde{c}, n) = \left( \sum_{b \in I_n} s(\tilde{c}, b) \right) - \lambda \left( \max_{r \in R} s(\tilde{c}, r) \right),$$

(2)

where $\lambda$ is configurable parameter.

In addition, they introduce another method to score clues not only on the basis of word similarities, but also on the basis of their frequency and the similarity of Dict2vec vectors (Tissier et al., 2017) – but this is actually a modification of the original distance matrix.

Their results show that relatedness calculated by GloVe performs best in combination with dictionary definitions and frequency, but without the latter, cosine similarity in FastText proves to be the best measure.

Furthermore, Kumar et al. (2021) studied if the decisions of human players can be predicted in an amended version of Codenames. For the predictions, they used word2vec and GloVe word embeddings, as well as several similarity measures on free association datasets, in particular SWOW (De Deyne et al., 2019) and USF (Nelson et al., 2004). They found that similarity based on random walks in SWOW performed the best, from which they concluded that not only direct associations, but indirect connections are also important in this game.

## 4 Our Codenames Agents

Building on the studies of Spence and Owens (1990), we introduce several word relatedness measures based on co-occurrences, which we expect to be more suitable for modeling the human perception of word connections than representation methods created for other NLP tasks. We create spymaster agents with several new clue scoring functions combined to our relatedness measures. This way, our methods only require a raw text corpus of appropriate size, so they can be used for any language. We evaluate them in two languages (English and Hungarian), in an online game with human players.[1]

### 4.1 Relatedness measures

Considering the previous results on the relationship between associations and co-occurrences (Spence and Owens, 1990; Shen et al., 2018), we create our distance matrices not from the latest neural methods of NLP, but from co-occurrences counted

---

[1] The game:
http://spymasters.herokuapp.com/
Source code and data:
https://github.com/xerevity/
CodeNamesAgent

in raw text. As English corpora we use the concatenation of the English Wikipedia and the English OpenSubtitles corpus, consisting of 5.692 billion tokens in total. For Hungarian, we use the lemmatized version of the Hungarian Webcorpus (Nemeskey, 2020), also including the Hungarian Wikipedia (1.414 billion tokens). We work with vocabulary sizes 15K in English and 10K in Hungarian, and remove stopwords.

#### 4.1.1 FastText

Among the similarity measures of Koyyalagunta et al. (2021), generally FastText seems to be the best model. So, following the cited work, we create a relatedness matrix based on the cosine similarity of FastText vectors. That is, if $\mathbf{v}_i, \mathbf{v}_j$ are vectors corresponding to words $w_i, w_j$, then

$$s_F(w_i, w_j) = \cos(\mathbf{v}_i, \mathbf{v}_j).$$

For comparability with the other methods, we train our FastText models on the above corpora for English and Hungarian in 300 dimensions, using window size 10.

#### 4.1.2 Normalized PMI

A standard and probably the most common method to calculate word relatedness from co-occurrences is computing the pointwise mutual information (PMI) of two words. However, PMI has well-known shortcomings, such as overvaluing the relatedness of rare words, and lacking a fixed upper and lower bound. Bouma (2009) introduced normalized PMI as

$$\text{PMI}_{\text{norm}}(x, y) = \left( \ln \frac{p(x, y)}{p(x)p(y)} \right) \Big/ -\ln p(x, y),$$
(3)

which has 1 and $-1$ as upper and lower bounds, and works well empirically as an association measure. According to a known practice, we keep positive values only.

Comparing this relatedness measure to data obtained from humans (MEN, Bruni et al., 2012 and WS-353 relatedness, Agirre et al., 2009), we found that taking the square root of $\text{PMI}_{\text{norm}}$ increases the Pearson correlation coefficient between human annotations and our calculated relatedness from 0.72 to 0.76 for MEN, and from 0.57 to 0.63 for WS-353. Additionally, in our following methods, it is beneficial if the values do not concentrate around zero, therefore we use the square root of normalized PMI hereinafter:

$$\text{NPMI}(x, y) = \sqrt{\text{PMI}_{\text{norm}}(x, y)}. \quad (4)$$

### 4.1.3 Squared NPMI matrix

In Codenames, to get ahead in the game, spymasters have to give clues that are connected to many words that are probably unconnected to each other. As Kumar et al. (2021) showed, they might associate words that are not in a strong direct connection, but are only indirectly related (e.g. religion is not related to tree, but both are related to Christmas, therefore religion could be a clue for tree).

To model such indirect connections, we multiply the relatedness matrix by itself, and use the values of the squared matrix $S'$ as the relatedness measure between two words. By the definition of matrix multiplication,

$$S'_{ij} = \sum_{k=1}^{n} s_{i,k} \cdot s_{k,j},$$

that is, if we define $G_0$ as a graph whose neighborhood matrix is the NPMI matrix then $S'_{ij}$ is the sum of the product of the weights on all two-length paths $v_i - v_k - v_j$ in $G_0$. Since all edge weights are between 0 and 1, considering the weight of a path as the product of its edge weights gives a valid relatedness measure: longer paths and paths that contain smaller weights will yield to smaller relatedness values.

Artetxe et al. (2018) also showed on word embeddings, that different powers of embedding matrices are beneficial for word similarity and word relatedness tasks, and that the optimal power is higher for relatedness than for similarity.

Another advantage of this method is, that it reduces the number of zeros in the matrix. This is most important in the case of a guesser agent, because if the matrix consists of many zero values, some clues may not have any related words on the board according to our relatedness measure. However, if we have a nonzero value for all board words, we can take the relatedness between the clue word and the bad words into account, which might be beneficial for a spymaster agent as well.

### 4.1.4 NPMI graph

In the method described above, we already used a relatedness measure based on a graph constructed from NPMI values, where the weight of a path was the product of the weights of the edges on the path. This way, a greater value of edge or path weights corresponds to a stronger connection between the nodes. However, a more common way is that edge weights represent distance, and path

|          | NPMI  | NPMI$^2$ | Graph | FastText |
|----------|-------|----------|-------|----------|
| **NPMI**     |       | 0.495    | 0.820 | 0.393    |
| **NPMI$^2$** | 0.349 |          | 0.578 | 0.621    |
| **Graph**    | 0.442 | 0.602    |       | 0.427    |
| **FastText** | 0.295 | 0.524    | 0.319 |          |

Table 1: Pearson (upper tringle) and Spearman (lower triangle) correlation coefficients between our relatedness measures.

weights are the sum of the edges, so that stronger connections belong to smaller path weights. Since our NPMI values are between 0 and 1, we can define graph $G$ as follows: an edge $e(v_1, v_2)$ between vertices corresponding to words $w_1$ and $w_2$ exists if and only if $\text{NPMI}(w_1, w_2) > 0$, and its weight is $w(e(v_1, v_2)) = 1 - \text{NPMI}(w_1, w_2)$. Now the distance between $w_1$ and $w_2$ is given by the weight of the shortest path between $v_1$ and $v_2$:

$$d_G(w_i, w_j) = \min_{\pi \in \Pi_G(v_i, v_j)} \sum_{e_k \in \pi} w(e_k), \quad (5)$$

We can turn these distance values into relatedness measures by subtracting them from 1:

$$s_G(w_1, w_2) = 1 - d_G(w_i, w_j). \quad (6)$$

This way, for two strongly related words, for which the shortest path is the edge between them, we get the NPMI as relatedness value. This method therefore has some of the advantageous properties of both above relatedness measures.

### 4.1.5 Comparison and evaluation of relatedness measures

To investigate the relationship of the above defined relatedness measures, we compute correlations between the score they assign to 100.000 random word pairs. As Table 1 shows, none of the measures are near equivalent, but they have nonzero correlations. They also show high positive correlations with MEN (Bruni et al., 2012) and WS-353 relatedness (Agirre et al., 2009), as can be seen in Table 2, which is hopeful for their usability as relatedness in Codenames agents.

### 4.2 Clue scoring functions

Say that the agent plays in the blue team, i.e. we want to generate clues associated to the blue words, based on the distance functions above. The functions of Kim et al. (2019) (see (1)) determined the score of a possible reference based on relatedness

| | MEN | | WS-353 | |
|---|---|---|---|---|
| | Pearson | Spearman | Pearson | Spearman |
| **NPMI** | **0.761** | **0.749** | 0.632 | **0.649** |
| **NPMI**$^2$ | 0.627 | 0.670 | 0.502 | 0.545 |
| **Graph** | 0.754 | 0.735 | **0.650** | 0.647 |
| **FastText** | 0.732 | 0.737 | 0.562 | 0.564 |

Table 2: Correlation between our relatedness measures and gold standard annotations.

of the clue word to the least related blue word targeted. The shortcoming of this, however, is that in addition to blue (good) words that are similar to the clue word, there may be bad words of a different color that are only very slightly less similar to the clue. We can assume that in this case, agents are less likely to choose the targeted words; or in general, the smaller the difference between the distances of two words from the clue according to our distance function, the more likely the human player will perceive the order of the two words reversed.

To avoid such problems, Koyyalagunta et al. (2021) (see (2)) add a penalty on the relatedness of the closest bad word to their scoring functions. This scoring function generally improves the quality of the generated clues, thus we use this as one of our scoring functions. However, this function does not require all bad words to be less similar to the clue word than the targeted words, and in our experiments there have been such cases that this caused a problem. Therefore we define *KoyyRestrict*, a restricted modification of $g_{Koyy}$:

$$g_{KoyyR}(\tilde{c}, n) = \begin{cases} g_{Koyy}(\tilde{c}, n), \\ \quad \text{if } \min_{b \in I_n} s(\tilde{c}, b) > \max_{r \in R} s(\tilde{c}, r) \\ 0, \quad \text{otherwise.} \end{cases} \tag{7}$$

Another disadvantage of this scoring function is that the sum of the similarities might be high even if only one targeted word is very related to the clue word, and the scores of the other targets are close to the scores of the bad words. Regarding this, replacing the sum (which is, in optimization for a certain $n$, equivalent with the arithmetic mean) with the harmonic mean of the relatedness scores might also lead to an improvement, especially if there are outliers among the vocabulary words with very high relatedness to a blue word. Thus, we introduce *Harmonic* scoring function as:

$$g_H(\tilde{c}, n) = \begin{cases} H(b|b \in I_n) - \lambda \cdot \max_{r \in R} s(\tilde{c}, r), \\ \quad \text{if } \min_{b \in I_n} s(\tilde{c}, b) > \max_{r \in R} s(\tilde{c}, r) \\ 0, \quad \text{otherwise,} \end{cases} \tag{8}$$

where $H$ is the harmonic mean function:

$$H(x_1, x_2, \ldots, x_n) = \frac{n}{x_1^{-1} + x_2^{-1} + \cdots + x_n^{-1}}.$$

Finally, we also use a different version (*HarmonicDivide*) of the above, where the penalty on the bad words is performed as division instead of subtraction:

$$g_{HD}(\tilde{c}, n) = \frac{H(b|b \in I_n)}{\max(n \cdot \max_{r \in R} s(\tilde{c}, r), 1)}. \tag{9}$$

We combine these four scoring functions with all of the above relatedness measures, and evaluate the agents thus obtained in the next section.

## 5 Evaluation and Analysis

Following Koyyalagunta et al. (2021), we use $\lambda = 0.5$ for *Koyyalagunta* and *KoyyRestrict* scoring functions, but also for the *Harmonic* function. We pair all relatedness measures to all scoring functions, creating 16 agents in total, and generate clues for $n = 2$ and 3 targeted blue words using all of them. Differently from Koyyalagunta et al. (2021), we consider all of our vocabulary words as possible clue words. For each possible clue word, the best target words in the set $I_n$ are the $n$ closest words to the clue word, so scoring a possible clue is computationally inexpensive.

We randomly create 100 boards, with each containing 10 good and 10 bad words. For each board, we generate clues with the 32 configurations detailed above. This results in 1304 distinct clues in English, and 1399 in Hungarian. For evaluation, we create an online game, where human players get a board with one of the corresponding clues randomly, and have to choose the given number of words from the board which they think the clue refers to. The players do not know how the agents work, and to avoid that through the game they learn it at the end of the round they only see the color of their chosen words. We collected 443 rounds played in English, and 1365 in Hungarian. This way, we have 31.5 rounds on average to evaluate English configurations, and 64 rounds for Hungarian. For one board, players on average spent 39 seconds on guessing in English, while 37 seconds in Hungarian. We note that the players of the Hungarian game were most likely Hungarian native speakers, while the same cannot be said about the English game, therefore we consider the Hungarian data more reliable.

| Evaluation | Relatedness | Koyy | KoyyR | HM | HM-Div | Koyy | KoyyR | HM | HM-Div |
|---|---|---|---|---|---|---|---|---|---|
| | | | | 2 targets | | | | 3 targets | |
| P@all | FastText | 0.764 | 0.757 | 0.740 | **0.829** | 0.710 | 0.712 | 0.756 | **0.759** |
| | NPMI | 0.747 | 0.747 | 0.776 | 0.715 | 0.707 | 0.708 | 0.733 | 0.695 |
| | NPMI$^2$ | 0.722 | 0.742 | 0.725 | 0.744 | 0.666 | 0.696 | 0.746 | 0.729 |
| | Graph | 0.795 | 0.795 | 0.827 | 0.715 | 0.727 | 0.735 | **0.759** | 0.695 |
| P@targets | FastText | 0.558 | 0.567 | 0.581 | **0.625** | 0.531 | 0.518 | **0.585** | 0.582 |
| | NPMI | 0.504 | 0.504 | 0.519 | 0.546 | 0.515 | 0.513 | 0.503 | 0.495 |
| | NPMI$^2$ | 0.529 | 0.547 | 0.554 | 0.479 | 0.503 | 0.513 | 0.556 | 0.550 |
| | Graph | 0.533 | 0.533 | 0.574 | 0.546 | 0.541 | 0.542 | 0.511 | 0.495 |

Table 3: Rate of correct guesses made by human players in the Hungarian game. Numbers falling into the bootstrapped confidence interval of the best score are underlined in each category.

| Evaluation | Relatedness | Koyy | KoyyR | HM | HM-Div | Koyy | KoyyR | HM | HM-Div |
|---|---|---|---|---|---|---|---|---|---|
| | | | | 2 targets | | | | 3 targets | |
| P@all | FastText | 0.707 | 0.726 | **0.783** | 0.722 | 0.711 | 0.742 | 0.755 | 0.760 |
| | NPMI | 0.727 | 0.727 | 0.670 | 0.682 | **0.764** | **0.764** | 0.725 | 0.716 |
| | NPMI$^2$ | 0.611 | 0.583 | 0.604 | 0.729 | 0.645 | 0.583 | 0.638 | 0.649 |
| | Graph | 0.714 | 0.714 | 0.679 | 0.682 | 0.750 | 0.750 | 0.723 | 0.716 |
| P@targets | FastText | 0.487 | 0.535 | **0.581** | 0.555 | 0.549 | 0.495 | **0.577** | 0.520 |
| | NPMI | 0.420 | 0.420 | 0.397 | 0.426 | 0.549 | 0.549 | 0.541 | 0.508 |
| | NPMI$^2$ | 0.377 | 0.361 | 0.372 | 0.445 | 0.354 | 0.369 | 0.370 | 0.470 |
| | Graph | 0.392 | 0.392 | 0.384 | 0.426 | 0.552 | 0.552 | 0.533 | 0.508 |

Table 4: Rate of correct guesses made by human players in the English game. Numbers falling into the bootstrapped confidence interval of the best score are underlined in each category.

Similar to Koyyalagunta et al. (2021), we compute the precision of the agents as

$$\text{P@targets} = \frac{|I_n \cap U|}{n},$$

where $I_n$ is the set of the targeted words, and $U$ is the set of words chosen by the players. However, the scoring functions optimize clue words to stay away from red words, but not from non-targeted blue words, which might be almost as related to the clue as the targeted ones. If the user chooses such an untargeted word, the agent still performs well. So we define P@all,

$$\text{P@all} = \frac{|A \cap U|}{n},$$

where $A$ is the set of all good (blue) words. In Table 3 and Table 4, we show the mean precision of the players' guesses on the clues of each agent. In each category (defined by language, evaluation method, and the number of targets), we construct a 0.95 level confidence interval for the best mean

precision using bootstrap, and mark the numbers falling into this interval underlined.

Among the configurations, FastText similarity combined with the *Koyyalagunta* scoring function was evaluated previously by Koyyalagunta et al. (2021), where it was the best agent without any language-specific resource, i.e. using raw corpora only. The results show that this is outperformed by many of our new configurations.

On FastText relatedness, our *Harmonic* and *HarmonicDivide* scoring functions result in a substantial improvement. Most of the best performing configurations use FastText as similarity measure combined with these functions, although the advantage of these methods is less significant when the guesses are evaluated on all blue words instead of the targets of the agent. Also, the only agent that performs within the confidence interval of the best agent in their category is FastText combined with *HarmonicDivide*, therefore we consider it as our highest performing agent. The second best agents

in this regard, falling short in one category only, are the *Graph* similarity combined with *Koyyalagunta* and *KoyyalaguntaRestrict* functions.

As we can see, different relatedness measures fit different scoring functions. As mentioned in 4.2, we think that the *Harmonic* functions are more beneficial where outliers with high relatedness can be found; more generally, the optimal clue scoring function depends on the distribution of the relatedness measures. The exact connection between them seems to be an exciting direction for future work.

Interestingly, the correlations of the relatedness measures to human-annotated relatedness data (seen in 4.1.5) are not predictive of their performance in Codenames, as in those experiments Fast-Text had been outperformed by both NPMI and Graph relatedness. The results in the two languages are not perfectly in line either. For example, in English NPMI$^2$ and graph relatedness perform worse than the two other relatedness measures, while the same does not appear in Hungarian. We suspect that this is because NPMI$^2$ and graph relatedness capture more indirect connections, which are more problematic to see for non-native speakers.

## 6   Summary and Future Work

In this work, we separated the Codenames spymaster agent's task into two parts. To cooperate with humans, we first need to specify a relatedness matrix that sufficiently approximates the relationships as judged by humans, and then define a scoring function on top of this that ranks the possible clues according to how many good guesses a human player is expected to give.

Based on previous research on associations, we generated some of our relatedness matrices based on co-occurrences between words in a corpus. We evaluated these relatedness measures with human-annotated relatedness data. However, we found that these scores were not predictive of the performance of the Codenames agents based on these measures.

We also introduced innovations in terms of scoring functions, firstly by refining the scoring function of Koyyalagunta et al. (2021), and secondly by using the harmonic mean of the relatedness to the clue word. This improved the performance of the best agents substantially.

Our best agents overall were FastText cosine similarity combined with a function using harmonic mean, and path weights in a graph of co-occurrences, combined with functions using arithmetic mean of similarities. This raises the question about what relationship is there between relatedness and scoring functions.

In future work, we would like to collect data on human spymaster-player decisions and evaluate guesser agents on them, which will directly allow the optimization of the relatedness measure.

Although many NLP methods have already been used to generate distance matrices, others are worth trying. Examples include graph embedding of associations (Bel-Enguix, 2014) and GraphGlove (Ryabinin et al., 2020).

As each relatedness measure can be defined by a matrix, it is also possible to aggregate several matrices generated in different ways. For example, creating distance matrices based on co-occurrences, neural word representations, and knowledge graphs at the same time seems to be a promising new direction. The comparison of such different relatedness matrices could also provide important lessons in cognitive modeling and the interpretability of neural word representations.

## Acknowledgements

We thank the testers of our game for the data needed for the evaluation, and the reviewers for their helpful suggestions contributing to the final version of the article.

Réka Cserháti was supported by the ÚNKP-21-1 – New National Excellence Program of the Ministry for Innovation and Technology from the source of the National Research, Development and Innovation Fund. This study was partially supported by the Ministry of Innovation and the National Research, Development and Innovation Office within the framework of the Artificial Intelligence National Laboratory Programme. Project no. TKP2021-NVA-09 has been implemented with the support provided by the Ministry of Innovation and Technology of Hungary from the National Research, Development and Innovation Fund, financed under the TKP2021-NVA funding scheme.

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

# A   Appendix: Example clues

Figure 1 is a board we used for evaluation, and Table 5 contains the clues generated by all of our agents for this board.

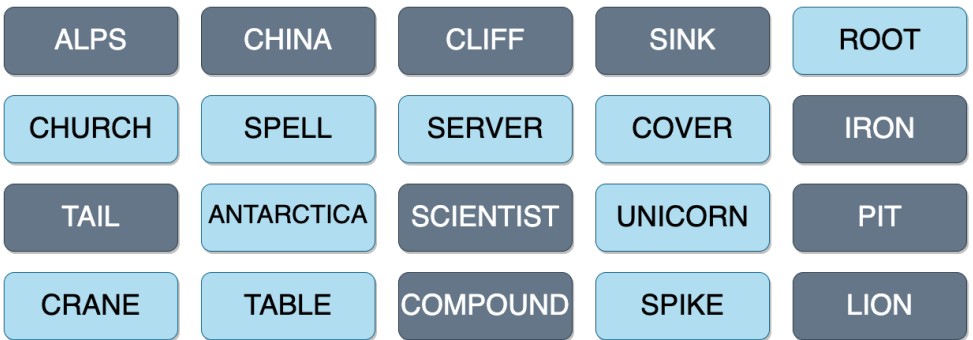

Figure 1: An example board used in evaluation

| Relatedness | Scoring | Number | Clue word | Target words |
|---|---|---|---|---|
| FastText | Koyyalagunta | 2 | chapel | church, crane |
| FastText | Koyyalagunta | 3 | raven | unicorn, crane, spike |
| FastText | KoyyRestrict | 2 | chapel | church, crane |
| FastText | KoyyRestrict | 3 | shark | unicorn, crane, spike |
| FastText | Harmonic | 2 | menu | table, server |
| FastText | Harmonic | 3 | bean | root, crane, spike |
| FastText | HarmonicDivide | 2 | doll | unicorn, spike |
| FastText | HarmonicDivide | 3 | preview | cover, server, spike |
| NPMI | Koyyalagunta | 2 | directory | root, server |
| NPMI | Koyyalagunta | 3 | altar | church, table, server |
| NPMI | KoyyRestrict | 2 | directory | root, server |
| NPMI | KoyyRestrict | 3 | altar | church, table, server |
| NPMI | Harmonic | 2 | directory | root, server |
| NPMI | Harmonic | 3 | altar | church, table, server |
| NPMI | HarmonicDivide | 2 | directory | root, server |
| NPMI | HarmonicDivide | 3 | altar | church, table, server |
| $NPMI^2$ | Koyyalagunta | 2 | user | server, root |
| $NPMI^2$ | Koyyalagunta | 3 | voiced | crane, spike, unicorn |
| $NPMI^2$ | KoyyRestrict | 2 | user | server, root |
| $NPMI^2$ | KoyyRestrict | 3 | voiced | crane, spike, unicorn |
| $NPMI^2$ | Harmonic | 2 | node | root, server |
| $NPMI^2$ | Harmonic | 3 | voiced | crane, spike, unicorn |
| $NPMI^2$ | HarmonicDivide | 2 | download | server, cover |
| $NPMI^2$ | HarmonicDivide | 3 | itunes | server, cover, unicorn |
| Graph | Koyyalagunta | 2 | directory | root, server |
| Graph | Koyyalagunta | 3 | directory | root, server, table |
| Graph | KoyyRestrict | 2 | directory | root, server |
| Graph | KoyyRestrict | 3 | directory | root, server, table |
| Graph | Harmonic | 2 | directory | root, server |
| Graph | Harmonic | 3 | altar | church, table, server |
| Graph | HarmonicDivide | 2 | directory | root, server |
| Graph | HarmonicDivide | 3 | altar | church, table, server |

Table 5: Clues generated for the board in Figure 1.