# OpenReview forum: "Codenames as a Game of Co-occurrence Counting"
_aclweb.org/ACL/2022/Workshop/CMCL — CMCL 2022_

### Official Review · Reviewer_zsee · 2022-03-25
**nice work**

**Rating:** 9
**Confidence:** 3

**Review:**

This paper builds on and improves previous work on modeling the Codename game. The authors introduce both new semantic association measures and new scoring functions that optimize the choice of clue words. The authors also compare the model predictions to human players online in both English and Hungarian.

The paper is well written,  the methods are clearly justified, and the results are compared with previous work and show increased performance.

I am not an expert in this sub-filed of study but I enjoyed reading the paper and found it interesting. I do not have any major concerns and thus, I recommend acceptance.

---

### Official Review · Reviewer_GDsD · 2022-03-25
**Solid work but some questions about conclusions and prior literature**

**Rating:** 6
**Confidence:** 4

**Review:**

This paper contributes to a small but growing literature investigating the codenames board game as a way to evaluate models of human semantic memory. The authors separate sources of semantic relatedness and decision rules and evaluate both of these in their ability to guide human players in an online game.

The direct evaluation of algorithm outputs is a major strength of the paper, as is the systematic evaluation of the space of different relatedness/scoring pairs. There is surprisingly little variation in performance among relatedness/scoring pairs, however, and there  is not much systematicity (e.g., sometimes a particular scoring rule performs better with one relatedness measure and worse with another). Further,  no statistical analysis is presented. These observations make me worried that – despite the large amount of data – the conclusions in favor of specific settings are tentative.

A notable new line of work by Kumar and colleagues has investigated both other sources of relatedness and judgments and other metrics, though both the details of the games and the evaluations are different enough to be difficult to compare:
- Kumar et al. (2021a) - https://escholarship.org/uc/item/92m138t3
- Kumar et al. (2021b) - https://onlinelibrary.wiley.com/doi/full/10.1111/cogs.13053

Also, there was no reference to an older, foundational paper, Xu & Kemp (2010) - https://proceedings.neurips.cc/paper/2010/file/766d856ef1a6b02f93d894415e6bfa0e-Paper.pdf

In sum, this seems like solid work but my enthusiasm was dampened by my limited confidence in the meaningfulness of results comparisons and by the omission of prior work.

---

### Official Review · Reviewer_28kX · 2022-03-26
**Interesting work on improving Codenames clue generation but conclusions are unclear**

**Rating:** 8
**Confidence:** 3

**Review:**

This paper reports an attempt to improve models/agents playing the game Codenames. The authors use co-occurrence-based relatedness measures to select clues. They modify the scoring functions used in prior work and compare performance of all pairwise combinations of their relatedness measures and scoring functions at generating clues which lead to correct guesses by English and Hungarian speakers.

MAJOR

This is an interesting study though the implications for cognition or linguistics could be made more salient given the scope/audience of CMCL. The innovations in terms of scoring functions seem useful. However, I'm not quite sure what the main take-away of this study is. Performance across all the combinations was very different for English vs. Hungarian. The authors suggest that the Hungarian data are more valid, but it's unclear which differences are reliable/interesting.

I also found it odd that the choice of lambda isn't justified anywhere. It seems like it would be helpful to know more about how it impacts the agents' behavior, how the value of 0.5 was chosen in previous work, and whether that is still the optimal value given the new relatedness metrics and scoring functions.


MINOR

The abstract of the paper is not very clear. I think it assumes that the reader already knows how codenames works. I actually did and still found it very opaque.

The paper fails to cite relevant work by Kumar, Garg, and Hawkins (2021). It would be useful to reference this work and clarify any major differences between their approach and the one presented here.

Ashok Kumar, A., Garg, K., & Hawkins, R. (2021). Contextual Flexibility Guides Communication in a Cooperative Language Game. Proceedings of the Annual Meeting of the Cognitive Science Society, 43. Retrieved from https://escholarship.org/uc/item/92m138t3

---

### Decision · Program_Chairs · 2022-03-29

Accept